# Exploring Perceptions of the Work Environment among Psychiatric Nursing Staff in France: A Qualitative Study Using Hierarchical Clustering Methods

**DOI:** 10.3390/ijerph17010142

**Published:** 2019-12-24

**Authors:** Baptiste Cougot, Ghozlane Fleury-Bahi, Jules Gauvin, Anne Armant, Paolo Durando, Guglielmo Dini, Nicolas Gillet, Leila Moret, Dominique Tripodi

**Affiliations:** 1Department of Occupational Medicine and Environmental Health, Work and Health Innovation Research Laboratory, Nantes University Hospital, F 44093 Nantes, France; jules.gauvin@chu-nantes.fr (J.G.); anne.armant@chu-nantes.fr (A.A.); 2EE1901 QualiPsy, University of Tours, F 37 000 Tours, France; nicolas.gillet@univ-tours.fr; 3Psychology Laboratory of Pays de la Loire LPPL - EA 4638, Department of Psychology, University of Nantes, F 44 000 Nantes, France; ghozlane.fleury-bahi@univ-nantes.fr; 4Department of Health Sciences (DISSAL), Occupational Medicine, University of Genoa, 16132 Genoa, Italy; durando@unige.it (P.D.); guglielmo.dini@unige.it (G.D.); 5Occupational Medicine Unit, Policlinico San Martino Hospital, 16132 Genoa, Italy; 6UMR INSERM U 1246–EA 4275–Methods in Patients-centered outcomes and Health Research-SPHERE, 22 Bd Bénoni Goullin, University of Nantes, F 42 200 Nantes, France; leila.moret@chu-nantes.fr

**Keywords:** qualitative, hierarchical, clustering, psychiatric, nurse, occupational, health, stress, perception, hospital

## Abstract

Most studies on workers’ health are based on non-specific models of occupational stress, thereby limiting the understanding and research on efficient interventions. This qualitative approach aimed to explore the structure of resources and constraints in the working environment of nurses in a deliberately open approach. Semi-structured interviews were conducted with 37 nurses working in closed and open inpatient psychiatric wards in a French university hospital. The data were statistically analyzed using a hierarchical clustering method. Our model highlighted a systemic structure, describing the interactions, including patients, nurses, doctors, and managers in a specific material, communicational, and organizational environment. The results show a discursive structure organized around dimensions pertaining to “environment”, “patients”, “medical-care group”, and “the individual”. Our model showed interest in an interdisciplinary approach that encompasses occupational medicine and social psychology.

## 1. Introduction

Many models have explained stress at work, the most common being the Job Demand, Decision Latitude, and Mental Strain model, the Effort-Reward Imbalance model, the Organizational Justice model, and the Leadership model [1,2,3,4,5,6].

Changes in work organization and workplace conditions have created new occupational hazards and risks for workers, with new and emerging challenges in the management for occupational health and safety stakeholders. A large body of literature has recognized several job stressors (e.g., long working hours, reporting to multiple employers, bullying and harassment, shift work, effort-reward imbalance, workforce shortage) combined with individual factors (e.g., gender, age) and other risks beyond the workplace (e.g., environment, work-life conflict) that could lead to mental health, depression [7,8,9,10,11], cardiovascular diseases, and sickness absence [12,13,14,15,16]. 

In particular, work-related stress arises when coping strategies are insufficient with respect to workplace demands, based on each individual worker’s knowledge and skills [17]. 

According to the current literature, it is estimated that between 20% to 30% of workers are exposed to work-related stress in Europe, with a higher prevalence in the USA (40%) and Canada (60%) [18,19].

Work-related stress can lead to mental health impairment as well as other physical disorders. It can contribute to the development of anxiety, depression, subclinical psychological distress, or other syndromes, such as burnout [20,21]. Work-related stress is most prevalent in the healthcare sector, with higher levels associated with compromised patient care [22]. Lazarus and Folkman [23] theorized that coping could be categorized as either problem-focused (e.g., by employing problem solving and time management strategies) or emotion-focused (e.g., through mindfulness, relaxation, and obtaining emotional support from colleagues or friends). In the context of healthcare, the ability to cope with stressors has been linked to improved mental health [24,25]. 

Although general nursing is recognized as a highly stressful occupation, the risk of psychological disorders is more increased among nurses working in psychiatric wards and hospitals [26,27,28], likely caused by the fact that, at least partly, in addition to stressors shared with general nurses, psychiatric nurses are often exposed to potential suicides, physically threatening patients, verbal abuse, and inadequate staffing [29,30,31]. These factors could affect work activities through labor turnover, absenteeism, poor morale, and reduced performance [32,33], and consequently, the quality of care provided by hospitals. Therefore, it is important to investigate and assess healthcare workers’ exposure to several job stressors. The visibility and recognition of stress among nurses has increased in the literature during recent years [34].

The value of statistically proven psychosocial and organizational models [1,2,3,4,5,6,23,35,36,37,38] developed to explain the worker-workplace interaction among health professions, including psychiatry, is well established [39]. These generic models are used for all kinds of activities to model the factors involved in stress, perceived health, or burnout. Many researchers from different countries have used these models [39,40]. Although these models apply to different activities, they are not specific to a population and may, therefore, be insufficient to describe all health determinants in psychiatry [41]. Moreover, the challenge goes beyond the scientific interest and concerns the foundation and efficiency of the intervention to improve working conditions. As Burns stated in response to Johnson’s survey, “it would be a shame if the debate were restricted to the framework of the demand-support-control model” [39,42]. That is why the revision of the Nursing Work Index considered nurses within a professional practice environment and explained differences in nurses’ stress levels and risk of burnout [43]. In the psychiatric field, Prosser et al. provided specific tools for psychiatric constraints [44]. Rosberg JI et al. considered the attitude of nurses or the role of “ward atmosphere” as a moderator [45]. Most of those approaches were quantitative, and few qualitative approaches are referenced today. In order to improve in this regard, the aim of our survey was to explore nurses’ constraints based on a statistical categorization of discourse, according to Hagen et al. [46]. Therefore, the research question was: what are the factors that lead well-being and/or stress in a psychiatric ward in order to implement a new understanding model ?

## 2. Methods

A qualitative study was conducted performing a hierarchical cluster analysis of the text in order to conceptualize nurses’ interactions with all actors (e.g., managers, and physicians) in an organizational and communicational environment of a French university hospital department.

### 2.1. Setting and Population

In France, psychiatric healthcare is organized into units providing complete psychiatric care to the population of the same geographical area. Each unit provides at least one outpatient ward (medical-psychological center) and two inpatient wards; one is open and the other is closed. Closed wards are secured and designed for patients who refuse care, whereas open wards allow free movement and treat patients who are more stable. 

The study population consisted of the nursing staff attending at the psychiatric department of a French university hospital in Nantes, Pays de la Loire. 

### 2.2. Study Period and Sampling

The survey was performed in two periods: the first one between March and June 2017 and the second one between March and June 2018. In the first period, all nurses from closed inpatient wards were recruited, and in the second period, all nurses from open inpatient wards were recruited. All participants were enrolled by the occupational physician after the periodic clinical exam performed in the occupational health department. The only exclusion criterion was the refusal of a nurse to participate in the study.

### 2.3. Data Collection 

Semi-structured interviews were conducted in the Health and Occupational Medicine department provided by the occupational physician upon the agreement of the participants. On average, the interviews lasted for 38 minutes, with a standard deviation of 12 minutes. The purpose and context of the interview, including its recording and anonymity, were explained in a standardized way. Fours general questions were asked: “Can you tell me how you perceive the work in psychiatry? What does working in psychiatry mean to you?”. And then: “indicate, in your opinion, what are the main resources at your workplace? “Indicate, in your opinion, what are the main constraints at your workplace?”. The interviewer encouraged the participants to speak freely by rephrasing any questions that were not answered sufficiently. If the participant still struggled, a second question was asked: “How do you feel every day at work?”. All interviews were audio recorded and transcribed verbatims.

### 2.4. Data Analysis and Use of IRaMuTeQ Software

All interviews were transcribed, and analysis was performed using «*R Interface for Multidimensional analysis of Texts and Questionnaires»* (IRaMuTeQ) software (Pierre Ratinaud, LERASS, Toulouse, France).

This program allows different processing and statistical analysis of texts. It is based on the R Software (R Core Team, Vienna, Austria), and it is coded in the python programming language. In 2009, Pierre Ratinaud developed it for the French dictionary, but in recent years, many other full dictionaries in other languages have become available.

The IRaMuTeQ software provides five types of analysis: classic text statistics, specificities of research groups, descending hierarchical classification, similitude analysis, and word cloud [47].

This program employs the ascendant and descendant hierarchical clustering method on discourse. Based on the idea that the discourse of individuals is structured into “lexical worlds,” the data were, first, cut into “Text Segments” (TS) of a few sentences, and then successive correspondence factorial analyses (CFAs) were conducted on TS, in columns, and words, in lines.

From the first iteration, TS were ordered according to their factorial saturation and then grouped into two classes, in such a way that the gap between the classes (inter-class inertia) was maximal. Then a new CFA was performed on the class that explains more variance, producing a new TS grouping into two classes (making a total of three classes); this process was repeated on other classes [48]. In this way, the analyses identified lexical associations and structured the discourse into classes that gathered the associated TS. Thus, classes could be interpreted not only through the interpretation of their TS but also by analyzing the strength of the association between words and TS classes using a χ² test. After reading the TSs, represented by the participants’ answers and arranged by the software in text segments, we chose, as a criterion of analysis, the use of words that presented a larger chi-square and a *p* < 0.0001 for determining the binding force between them. Contrary to principal component analysis, the classes obtained must be considered interdependent and seen in terms of their opposition and proximity, as shown by the hierarchical structure.

### 2.5. Ethics and Informed Consent

The purpose and nature of the study was disclosed to all participants during the introduction to the study, and informed consent was obtained. They were informed that at any time, they could withdraw from the study without giving any reason. All interviews were audio recorded for later transcription. The study was approved by the Institutional Review Board prior to its start, and guidelines for the ethical conduct of research were upheld throughout the research process. This survey was conducted within the activities of a grant (research grant: program n° 09/118) from the Department of Occupational Medicine and Environment Health of Nantes University Hospital, *HCWs Research Lab., Work and Health Innovation Unit,* and received the Ethics Committee approval.

## 3. Results

This study included 37 nurses (Table 1), with a participation rate of 60%, mean age was 37.8 years, SD = 12.2 years; 81% were female; 19% were night shift workers; mean seniority in the unit was 5.3 years. The distribution was as follows: open wards, 17; closed wards, 16; and “s” the replacement team (requested to replace absent colleagues), 4.

The 33 nurses that worked in the open and closed units were distributed in five psychiatric units A, B, C, D, E: 6 in A; 10 in B; 5 in C; 7 in D; and 5 in E.

The final “verbatim corpus” comprised 37 texts, cut into 4213 Text Segments (TS), which, in turn, comprised 7104 word forms. Active word forms (nouns, verbs, adjectives) totaled 4227, making up 151,489 occurrences. Although the analysis was automatic, the number of generated classes was created as follows: (a) the number of classes was increased until the categories were qualitatively specific and homogeneous; (b) the classes included at least 5% of the total TS, and (c) the total TS classified remained above 90% of the whole corpus of TS. This resulted in a 10-class structure, which categorized 90.91% of the TS. A sample of text segments for each class can be found in Table 2.

### 3.1. Subsection

#### 3.1.1. Definition of Classes by Theme and Hierarchical Structure

The different classes by theme are reported in Table 3 with the percentage of text segment (TS), which categorize each class. 

The textual corpus was categorized into four themes, and 10 subclasses:

#### 3.1.2. Theme 1 

Theme 1 comprised a single class: ***Class 10*** (5.7% of the discourse) and clustering terms related to “environmental characteristics”, patient comfort, and nursing work. Words referred to “bedroom”, “bed”, “toilet”, “bath”, and “smoking room”. Other words, such as “horror”, “old”, and “decrepit”, referred to the ancient and dilapidated appearance of certain places. There were words related to nursing work, such as “(to) secure”, “(to) allow”, and “care”, and forms referring to staff amenities and comfort, such as “break”.

Regarding the environment related to the patient, nursing work, and staff comfort, generic terms, such as “room” and “place”. Other forms, such as “treatment room” and “bathroom” (separated in French), were associated with patients and care.

#### 3.1.3. Theme 2 (3 Classes/32.6% of TS)

***Class 7*** (12.1% of TS) is related to “patient”, his paroxysmal manifestations in terms of “violence”, and associated “pathologies” or “diseases” to “psychosis”. Other forms referred to specific features of the wards, such as “closed” and “open”, reflecting the differing types of hospitalization in relation to violence.

***Class 2*** (14.7% of TS) is related to the institutional care of patients who risk “running away” and “acting out”, in (and out of) the “intensive care room”, including the organizational and group support that the latter requires: “(to) call,” “backup,” and “alone worker protection.” Other forms referred to the “risk” associated with “(to) hurt”, or being “overwhelmed”.

***Class 3*** (5.8% of TS) is related to patient “treatment”, regarding “anxiety” or “(to) sleep”. Other terms were “night”, “evening”, and “daytime”, which reflected temporality a discursive opposition between day and night work. They were associated with TS, describing specific features of night work, from the viewpoint of care and certain difficulties related to working rhythm and communication with dayshift personnel.

Theme 2 included three classes related to the patient, including aggressive ones. Classes 2/3 were extremely close; they referred to institutional medical treatment by staff via the intensive care room and chemical treatment of less violent manifestations, respectively. These treatment-related classes were linked to Class 7, which is more centered on patients’ diseases and violence.

#### 3.1.4. Theme 3 (4 Classes/39.7% of TS)

***Class 1*** (16.7% of TS) describes the “team”, “the patient management”, interactions with the “manager” and “hierarchy”. Other forms were related to the way in which the team operates, in terms of “meeting”, “organization”, defining each “distinctive role”.

***Class 6*** (5.0% of TS) contains “everyday”, “communication”, “exchange”, or “discussion” between “staff”. It also contains the word “together” with a relatively high χ² value (rank 3), introducing the idea of group cohesion.

***Class 8*** (11.6% of TS) contains discourses on “doctor(s)” and doctor-nurse collaboration. Especially important are the ways in which, among others, a “decision” was made, the place given to the nurses’ “reference”, the doctors’ “presence” in the ward, the “time” “taken” to “reflect”, and the issues in terms of “functioning” or professional “interest”.

***Class 9*** (6.4% of the TS) referred to “schedule” problems, involving the management of “holidays,” “sick leave,” “changes,” and “replacements.”

Thus, Classes 8/1/6 “opposed” doctor–nurse and team–manager collaboration in interaction with the organization. Finally, there was a terminal distinction between team–manager–support–organization and communication–staff cohesion.

#### 3.1.5. Theme 4 (2 Classes/21.9% of TS)

***Class 5*** (9.5% of TS) concerns “age” and “young” nurses who recently left “school” to work in a psychiatric ward, regarding their “experience” and “skill” level. These elements described a lack of self-confidence, seeking refuge in protocol, contributing to making young nurses vulnerable in psychiatry, in contrast to having long experience.

***Class 4*** (12.4% of TS) referred to “professional life” and its link to “private life” (χ² = 115.8, *p* < 0.001). The specific TS shows more precisely the individual’s experience of their “job,” including its perceived impact on their private life and the need to preserve the boundary between work and private life.

In summary, Theme 4 includes the characteristics of the nurses in terms of their competency, experience, seniority, and involvement in their work, and the impact on their life, depending on their ability to maintain the boundary between their private and professional lives.

## 4. Discussion

The originality of this study lies in its statistical approach of the hierarchical classification of discourse. It provides a categorical structure with four themes and 10 subclasses, excluding the bias of confirmed expectations associated with a non-statistical analysis, thereby enhancing the strength of the results. Text analysis using IRaMuTeQ has been rarely performed previously in the occupational context. We only found five papers using IRaMuTeQ for the analysis of professional practices. Broc G et al. carried out a qualitative analysis of the significant factors collected in the studies reviewed about surgery practices. Five categories of factors (i.e., patient, surgeon, treatment, tumor, and organizational cues) were found to influence surgeons’ decision-making [49]. Gutierres LS et al. studied practices for patient safety in the operating room: they found eight nurses’ recommendations: involvement of the multi-professional team and the managers; establishment of a patient safety culture; use of the safe surgery checklist; improvement of interpersonal communication; expansion of nurses’ performance; adequate availability of physical, material, and human resources; individual search for professional updating; and development of continuing education actions [50]. Lowen IMV et al. analyzed the reorganization of the health care practice of nurses interviewing 32 management and care nurses. They found four subclasses: reorganization of schedules, nursing consultation, physical restructuring, and shared consultation [51]. Galindo Neto NM et al. studied teachers’ beliefs about first aid at primary and elementary schools. Three classes were obtained: teachers’ knowledge about first aid; feelings in situations of urgency and emergency; first aid at school [52]. In those studies, researchers focused on good practices and the management of work rather than the scope of mental health of workers. Nevertheless, we could find some similarities with our findings. The methodology using hierarchical classification highlights classes and subclasses, with possible implementation of recommendations. Indeed, Fernandes MA et al.conducted a descriptive-exploratory study with a qualitative approach with 10 patients under treatment at a psychiatric hospital in Northeastern Brazil to investigate their perception of the relationship between their illness and job. The textual corpus was categorized into six classes taking four aspects into account: characterization of participants, occupational data, health data, and medical treatment [53]. Notwithstanding the differences with our study, several similarities are present in the analysis of the text corpus, in particular, in the relationship between the classes concerning interpersonal relationships at work, perceived inability to achieve goals, dissatisfaction with work, and lack of flexibility and support at the workplace. Our analysis showed a discursive structure designed around four themes “ergonomic and work environment”, “patients”, “medical-care group”, and “needs of the whole person”.

### 4.1. Ergonomics and Work Environment

Regarding the environment, a link between the discourses related to where the patients are recovered, the type of care they receive, and nursing staff comfort is observed in the same Class. This result is consistent with the ward-atmosphere approach, which considered the environment as a representational object for both nurses and patients [45]. This, in turn, mediates the relationship between the system outcomes in terms of care quality and nurse satisfaction. Our results support the idea that this relationship has been integrated into the representation of the nurses in our sample. The fact that this class is superordinate to the patient, group, and individual taxa also supports this interpretation.

### 4.2. “Chemical and Institutional Care” for Patients

Patient-related Theme 2 shows the relationship between the patients and their pathological manifestations and therapeutic methods. The division of treatments into two classes (Classes 3 and 2), which are chemical and institutional care, reflects primarily the psychiatric care practices, which involved two separate actions.

In addition, it is conceivable that this distinction between chemical and institutional care shows the representational opposition between modern care and the legacy of “institutional therapy.” This paradigm largely guided French psychiatric practices at the end of the Second World War [54] and still permeates “the French institution” like a partially suppressed historic sediment” [55]. Classes 2 and 3 include elements related to the human, material, and spatial organization of care, as well as the risks for patients and nurses. This inclusion of risk provides information on how psychiatric care can be stressful, which is consistent with the literature that describes, for example, the effects of the threat of violence [56,57], the social level of the patient group [58], or face-to-face contact [40].

Moreover, the presence of risk, patient, and care in its organizational and collective components in the same class suggests the intricacy of all these parameters at a representational level. This result supports the interest of an integrated prevention for care quality and occupational health, which is now widely valued.

### 4.3. Collective: An Answer to Meet Work Demands

As mentioned previously, Theme 2 is connected directly to the group of Taxa 3 and 4, which describes the functions of a team and an individual’s ability to cope with work, respectively. Initially, this suggests an opposition between nurses (Theme 3 and 4) and patients (Theme 2) at the representational level. However, a further analysis demonstrates a link between the three taxa, with a manager–doctor–nurse–organization system designed to provide a solution to patient problems through collective work.

In fact, the term “taking charge/care” (a translation of the French *prise en charge*, which underlines the ideas of both care and collective responsibility) is specific for Classes 1 and 8 of Theme 3, which is related to the team–manager–organization and nurse–doctor–decision systems, respectively. This result is consistent with the literature, suggesting the importance of team processes for both occupational health and clinical performance [54].

### 4.4. Actors System and Processes of Collective Efficiency

To clarify the nature of these collective processes, the representational distinction between Classes 1, 6, and 8 in Theme 3 is interesting. Classes 1, 6, and 8 illustrate teamwork in care but with different emphases. Class 1 seems to highlight the positive or negative team-manager interaction, for the definition of an organization supporting communication and collective coping with situations. In contrast, Class 8 reflects more specifically the doctor–nurse interaction and the quality of collaboration in decision-making regarding the choice and implementation of care. There is a more organizational-communicational emphasis in the manager–nurse relationship and a therapeutic-decision emphasis in the doctor–nurse relationship, which is consistent with the function of these different professions.

Second, Class 6 confirms this specificity by associating the manager–team class with staff communication and cohesion. Although the factorial analysis does not allow for specific team processes to be modeled in terms of coordination, cohesion, specialization, or trust, it clearly isolates the discourse related to staff communication in a single Class, thus indicating the centrality of communication for collective efficiency in adjusting to situations in our sample.

### 4.5. Organizational Support

Still, in Theme 3, Class 9 appears superordinate to Classes 1, 6, and 8. Thus, it links the functioning of the actor system to scheduling problems. The issue is the challenge of managing staff working times, in consideration of absenteeism and its impact on nurses in terms of, among others, exhaustion.

In addition to local organization, this class involves a more institutional level, in terms of the resources (e.g., staff numbers, use of temporary staff, replacement staff, etc.) provided to units to cope with unexpected events and to limit their impact on team schedules. In summary, Class 9 links the actor system to organizational and structural support.

This representational relationship between collective functioning and organizational support is consistent with team efficiency in care [59] and with works on empowerment [60].

In these paradigms, the whole work environment is considered a matrix of resources and constraints, and of individual empowerment and collective intelligence, to increase efficiency and well-being [61].

### 4.6. Personal Coping, Impact on Personal Life

Directly linked to group-related Theme 3, Theme 4 clusters discussions on the individual characteristics of nurses, their personal ways of coping with work, and work’s impact on their private lives. These elements are consistent with the literature on psychology in the healthcare setting, showing the role of individual determinants, in terms of experience, skills, or personality, in explaining occupational health. At this level, age and experience appear particularly significant in Class 5, emphasizing their role in coping with work demands in our sample, which supports the literature in this field [39,40].

### 4.7. Overall Structure

Our study on the perceptions and values of nurses working in psychiatric units in their work environment reflects functional and systemic structure, describing, in an indirect manner, the interactions between all the actors, including patients, nurses, doctors, and managers, in a specific material, communicational, and organizational environment (Figure 1).

Therefore, rather than being a source of stress or satisfaction, the nurse–doctor–manager collective has an important place in the discourse and appears to be a central resource for coping with professional situations.

Despite the strengths in the methodological approach and the inferences derived from the results, our study showed some limitations. The first limitation in this study is surprisingly the participation rate of 60%; 40% of the nurses did not participate because of a lack of time during the work or because of absenteeism not substituted on the unit. Therefore, the issues contributing to this problem do not appear in the classes and theme. The second limitation is the modality of nurses’ inclusion: there were no randomization and no control group; this may warrant many cautions on the interpretation of the themes and their classifications. The third limitation was the fact nurses have to leave their workplace and come into the Health and Occupational Medicine department. Probably some of them did not participate because of fear-avoidance of clinical examination and/or interview.

This centrality of the collective and of communication is consistent with fundamental works in social psychology and is more applied to works in clinical performance [59]. In the laboratory, collective intelligence is the first predictor of group efficiency, followed by the average of individual IQs and the maximum individual IQ in the group. Moreover, this collective intelligence is best explained by the quality of communication, in terms of equity in speaking turns [62].

From this viewpoint, the key concern is the search for an organizational and psychosocial context, which is likely to promote communication, to construct and maintain an efficient collective at work. In other words, an efficient collective is one that can autonomously mobilize the resources needed to meet all work demands, including the use of environmental and organizational coping.

In this perspective, that is closer to empowerment [61] and team process models [63], conventional stress models have a more epidemiological value, which is less relevant to the operational promotion of nurses’ health and their work experience.

## 5. Conclusions

The use of a qualitative approach in exploring the work organization, as defined in Labor Law, is a pioneering study in France. This approach enables the implementation of a new model of occupational medicine observations and/or clinical examinations helping understand health care work, and to develop prevention plans and empowerment within the health care facility (Figure 2).

In addition, with a view of modeling occupational health in a medical department, our results suggest the benefits of exploring precisely and operating the psychosocial processes involved in the formation of efficient collectives, and the collectives’ relationship with the organizational and structural contexts, especially in terms of teams participating in defining the work resources at their disposal. This text mining approach can make occupational physicians aware of the most frequent constraints affecting workers’ perception and workforce satisfaction. On the one hand, this can be useful for an integration of risk assessment on the topic of psychosocial risks, during health surveillance activities, gaining information about emerging risk factors both at the workplace and individual levels. On the other hand, this can be useful for assessing the effectiveness of the workplace preventive measures in place. All this information can support insights orientating formation, preventive, and health promotion programs, aimed at enhancing workplace well-being tailored to individual work plans [64]. Occupational physicians, in collaboration with all occupational health and safety actors, including the management level, can modify or integrate ad-hoc preventive measures for mental health based on the specific organizational issues arisen from the analysis. Improving evidence-based occupational health surveillance programs is mandatory for workers health and safety protection in several occupational settings, including the Healthcare sector [65]. Our model showed interest in a poly-disciplinary approach that encompasses occupational medicine, social psychology, the psychology of work, and organizations. However, further research is necessary in order to fully evaluate the applicability and usability of this novel approach in health surveillance programs, with the aim of contributing to the management of mental health among workers.

## Figures and Tables

**Figure 1 ijerph-17-00142-f001:**
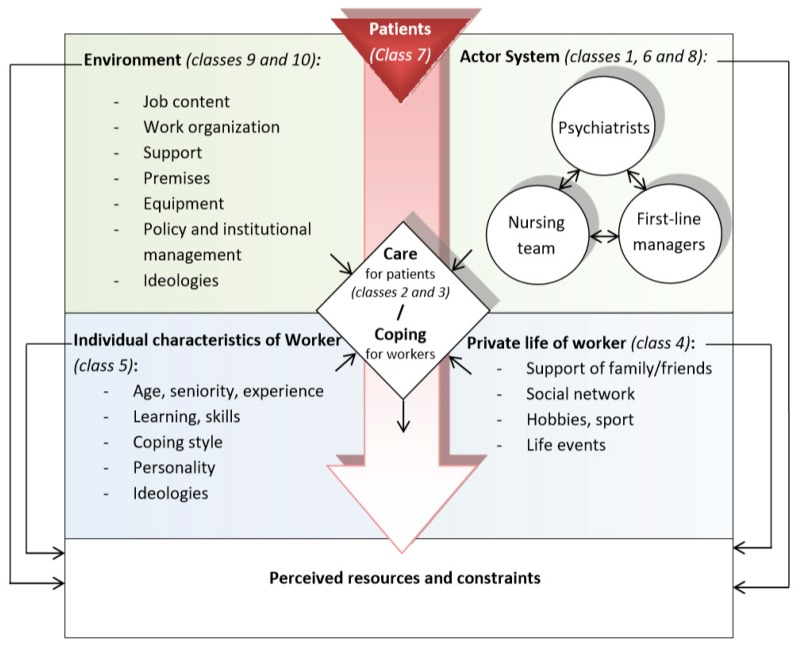
Hypothetical model of the relationship between classes in the generation of perceived resources and constraints.

**Figure 2 ijerph-17-00142-f002:**
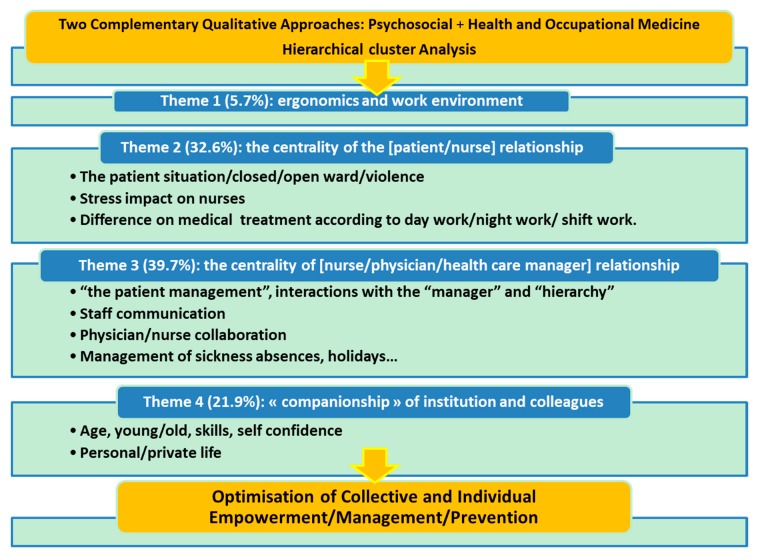
Conceptual map of Health Care Work Empowering.

**Table 1 ijerph-17-00142-t001:** Characteristics of the 37 participants.

Variables	Categories	*n*
*Age (years)*	23–29	11
	30–40	14
	41–65	12
*Gender*		
	Men	7
	Women	30
*Psychiatry sector*		
	a	6
	b	10
	c	5
	d	7
	e	5
	s	4
*Unit type*		
	Closed	16
	Open	17
	Variable	4
*Working time*		
	Day	30
	Night	7
*Seniority in unit (years)*		
	0–1	10
	2–4	10
	4–9	9
	10–28	7

**Table 2 ijerph-17-00142-t002:** Sample of specific text segments classified by class.

Class	Text Segments (TS)
Class 1	I asked my manager to set up a team meeting, because I wanted things to be said, in order to improve communication.
	In clinical meetings, we talk about patient care, and I’ve already seen managers talk about things they know nothing about.
	I expect the manager to be the nursing team spokesman, when things like that happen.
Class 2	I dodged his blows, I was lucky I wasn’t hit, and lots of colleagues came to help; the patient went to the CSI.
	When a patient who clashes arrives in the unit, we’re going to think about protecting the other patients, protecting the patient, to call for backup.
	An intensive care room is provided, we call enough men and then we accompany the patient to the room, usually we mustn’t touch the patient.
Class 3	You have to give treatments to patients who don’t need them, but as they can’t sleep because of another’s agitation.
	There are more anxieties in the evening so it’s always harder to manage, but patients are well managed during the daytime, so they are generally calm in the evening.
	A more sedative treatment that acts on anxiety, after a background treatment like risperidone… Then we mustn’t abrade too much as they’re not well afterwards.
Class 4	Outside it will be family, friends, spouse, sport, music … life goes beyond work so that’s important.
	I think that one day all psychiatric nurses need to go by themselves to verbalize things to a psychologist or a psychiatrist.
	In the evening I go home and I read a book or watch a series and it helps me to return to my private life and I manage better now.
Class 5	You’re a recent graduate, you’ve had a general training, you’re not trained in psychiatry and you begin in psychiatry … it all has an effect.
	You learn that with experience, the ability to protect yourself, you can’t learn that at school.
	As I’m a recent graduate, maybe I can’t step back, it’s experience that lets you stand back like my colleagues.
Class 6	He’ll have to listen to what you say, there’s an exchange that’s necessary; it’s more about communication than I tell you what to do.
	Our team gets along well, we communicate, there are exchanges, we can share our doubts, discuss.
	The more meetings we have, the more time we have together to exchange information directly.
Class 7	In a closed unit it’s really violence, or crisis, managing the crisis in the psychotic patient.
	It’s a serious disease, when psychotic delusional people are in an acute state of suffering, they don’t have access to reasoning.
	People who have behavioral disorders on the streets and go to prison from time to time … They have violent or even suicidal profiles.
Class 8	Relationship with doctors, I find it interesting when you have doctors who are there and who take the time with you, when nurses go to the medical interviews.
	Where there were no nursing interviews, where the word of the nurse was not listened to.
	The doctors, or the head of service, give us a lot of autonomy in decision-making.
Class 9	We make changes, we arrive on holiday we’re exhausted, and we tell ourselves that evening-morning, nights-days, ultimately it’s tiring.
	There are institutional demands for schedule change that don’t go with our work organization in psychiatry.
	It was a schedule that was quite difficult with many days of isolated rest.
Class10	So everyone has a bathroom, there are two rooms with 2 beds but with a bathroom so it’s still good.
	It’s quite scary I think to arrive in a place like that, such as intensive care rooms, it’s really the strict minimum.
	This patient, they took him out of the intensive care room, put him in a room although he’s still a man from prison and we had minors in the unit.

**Table 3 ijerph-17-00142-t003:**
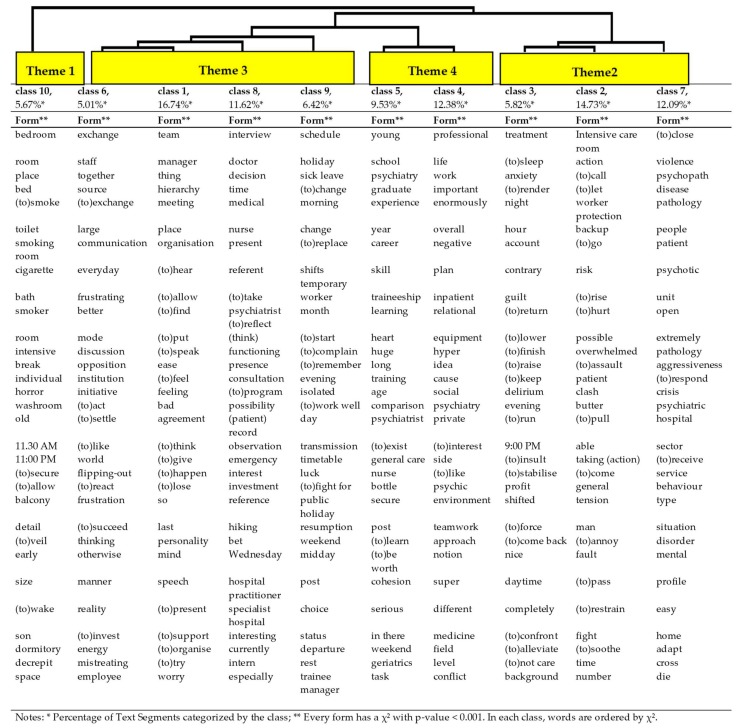
Hierarchical structure and words associated with classes.

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
