# Peer review of "Exploring Perceptions of the Work Environment among Psychiatric Nursing Staff in France: A Qualitative Study Using Hierarchical Clustering Methods"

_ijerph, 2019, doi:10.3390/ijerph17010142_

Round 1
Reviewer 1 Report
In the final part of the "Abstract" the authors should have presented the main dimensions or constructs of the model.
The review of literature should be more complete regarding the identification of job stressors. At the end of “section 1. Introduction” the authors should have written a research question.
In line 117 the authors only ask one question to measure the object and study - which is considered insufficient. Lines 121 and 122 have some French words. At line 169 - authors should be consistent in writing, such as table nº 2 or table 2
The results are well interpreted. However, the authors should define the model dimensions more clearly and the results should have been compared with other results from different authors, even from different contexts.
In section “5. Conclusions and implications for clinical practice”, the authors should have submitted a model proposal.
Author Response
Answer to Reviewer 1 IJERPH
Doctor Dominique Tripodi and all authors thank very much reviewer 1 for his helpful advice.
1° Comments and Suggestions for Authors
In the final part of the "Abstract" the authors should have presented the main dimensions or constructs of the model : we have mentioned the main dimensions of our model in the final part of the Abstract.
2° The review of literature should be more complete regarding the identification of job stressors.
Review of literature is implemented (lines 45-55). I added 4 models : the Job Demand, Decision Latitude, and Mental Strain model (R Karasek, 1979), the Effort-Reward Imbalance model (Siegrist, 1996), the Organizational Justice model ( Adams JS, 1963), the Leadership model (Lewin KLR, White RK, Lippitt R, 1939, 1960) [1-6].
Karasek, R. Job demands, job decision latitude, and mental strain: implications for job redesign. Adm. Sci. Quaterly 1979, 24, 285-308. Siegrist J. Adverse health effects of high-effort/low-reward conditions. J. Occup. Health Psychol. 1996, 1, 27-41. Adams JS. Toward an understanding of inequity. Journal of abnormal and social psychology 1963, 67, 422-36. Adams JS. Inequity in social exchange. Advances in experimental social psychology 1965, 2, 267-99. Lewin, KLR.; White, R.K. Patterns of aggressive behavior in experimentally created social climates. Journal of Social Psychology 1939, 10, 271-301. White, RK.; Lippitt, R. Autocracy and Democraty: An Experimental inquiry: Harper & Brothers; 1960.And work related diseases according to those models [7-16] :
Dewa, CS.; Lesage, A.; Goering, P.; Craveen M. Nature and prevalence of mental illness in the workplace. Healthc. Pap. 2004, 5, 12-25. Kivimaki, M.; Ferrie, JE.; Head, J.; Shipley, MJ.; Vahtera, J.; Marmot, MG. Organisational justice and change in justice as predictors of employee health: the Whitehall II study. J. Epidemiol. Community Health. 2004, 58, 931-7. Stansfeld, S.; Candy, B. Psychosocial work environment and mental health-a metaanalytic review. Scand. J. Work Environ. Health 2006, 32, 443-62. Bonde, JP. Psychosocial factors at work and risk of depression: a systematic review of the epidemiological evidence. Occup. Environ. Med. 2008, 65, 438-45. Nyberg, A. The impact of Managerial Leadership on Stress and Health Among Employees. Stockholm: Jarolinska institutet; 2009. Elovainio, M.; Kivimaki, M.; Puttonen, S.; Lindholm, H.; Pohjonen, T; Sinervo, T. Organisational injustice and impaired cardiovascular regulation among female employees. Occup. Environ. Med. 2006, 63, 141-4. Kivimaki, M.; Virtanen, M.; Elovainio, M.; Kouvonen, A.; Vaananen, A; Vahtera, J. Work stress in the etiology of coronary heart disease-a meta-analysis. Scand. J. Work Environ. Health 2006, 32, 431-42. Kivimaki, M.; Ferrie, JE.; Shipley, M.; Gimeno, D.; Elovainio, M.; de Vogli, R. et al. Effects on blood pressure do not explain the association between organizational justice and coronary heart disease in the Whitehall II study. Psychosom. Med. 2008 , 70, 1-6. Eller, NH.; Netterstrom, B.; Gyntelberg, F.; Kristensen, TS.; Nielsen, F.; Steptoe, A, et al. Work-related psychosocial factors and the development of ischemic heart disease: a systematic review. Cardiol. Rev. 2009, Mar, 17, 83-97. Kivimaki, M.; Head, J.; Ferrie, JE.; Singh-Manoux, A.; Westerlund, H.; Vahtera, J. et al. Sickness absence as a prognostic marker for common chronic conditions: analysis of mortality in the GAZEL study. Occup. Environ. Med. 2008, 65, 820-6.
3° At the end of “section 1. Introduction” the authors should have written a research question : the research questions is : what are the factors that lead well-being and/or stress in a psychiatric ward, line 89.
4° In line 117 the authors only ask one question to measure the object and study - which is considered insufficient. Indeed : Two others questions were asked : “what are ressources /constraints at your work place ?” : completed in the manuscript Lines 121,122.
5° Lines 121 and 122 have some French words : this has been modified.
6° At line 169 - authors should be consistent in writing, such as table nº 2 or table 2 : ok, this has been modified and clarified.
7° The results are well interpreted. However, the authors should define the model dimensions more clearly and the results should have been compared with other results from different authors, even from different contexts : we have compared with literature datas, only 5 papers are found using IRAMuteq for analysis of professional practices (see lines 245-277).
8° In section “5. Conclusions and implications for clinical practice”, the authors should have submitted a model proposal : we have created a conceptual map summerazing the 4 dimensions of our model and the possible effect on individual and collective empowerment : see Figure 1.
Doctor Dominique Tripodi and all authors thank very much reviewer 1 for his helpful advice.

Reviewer 2 Report
The paper is interesting, both in terms of contents and in terms of methodology. It is well written. it would improve addressing the following, minor points:
a) please clearly state the hypotheses or the research questions;
b) more literature review is needed on nurses'stess (p.2, line 68)
c) The Nursing Work index needs to be better introduced and linked to the previous paragraph;
d) Were the interviews individually administered?
e) p2, line 49 "comorbilidy" does not seem to be appropriate in a list of individual factors
f) authors did not mention the limits of the study;
g) authors did not elaborate sufficiently what their findings might offer for the management
Author Response
Anser to Reviewer 2
Doctor Dominique Tripodi and all authors thank very much reviewer 2 for his helpful advice.
The paper is interesting, both in terms of contents and in terms of methodology. It is well written. it would improve addressing the following, minor points:
a) please clearly state the hypotheses or the research questions : clarified line 89
b) more literature review is needed on nurses'stress (p.2, line 68) : we added lines (65-68):
“Lazarus and Folkman [23] theorised that coping could be categorized as either problem-focused
(e.g., by employing problem solving and time management strategies) or emotion-focused (e.g., through mindfulness, relaxation, and obtaining emotional support from colleagues or friends). In the context of healthcare, the ability to cope with stressors has been linked to improved mental health [24,25]”
Lazarus, R.S.; Folkman, S. Stress, Appraisal, and Coping; Springer Publishing Company: New York, NY, USA, 1984.
Chang, E.M.; Daly, J.W.; Hancock, K.M.; Bidewell, J.; Johnson, A.; Lambert, V.A.; Lambert, C.E. The relationships among workplace stressors, coping methods, demographic characteristics, and health in Australian nurses. J. Prof. Nurs. 2006, 22, 30–38.
Samaha, E.; Lal, S.; Samaha, N.; Wyndham, J. Psychological, lifestyle and coping contributors to chronic fatigue in shift-worker nurses. J. Adv. Nurs. 2007, 59, 221–232
c) The Nursing Work index needs to be better introduced and linked to the previous paragraph: changed with : “That’s why the revision of the Nursing Work Index considered nurses within a professional practice environment and explained differences in nurses’ stress level and risk of burnout”.
d) Were the interviews individually administered? In the Health and Occupational Medicine department (line 121).
e) p2, line 49 "comorbilidy" does not seem to be appropriate in a list of individual factors : “comorbilidy” has been removed.
f) authors did not mention the limits of the study; the limits are mentioned on 364-369 lines :
“Despite the strengths in the methodological approach and the inferences derived from the results, our study showed some limitations. The first limitation in this study is surprisingly the participation rate of 60%; 40% of the nurses didn’t participate because of a lack of time during the work or because of absenteeism not substituted on the unit. Therefore, the issues contributing to this problem do not appear on the classes and thema/taxa. The second limitation is the modalility of nurses inclusion: there were no randomization, and no control group; this may warrant many cautions on the interpretation of the thema and their classifications”. I added on line 370 : “The third limitation was the fact nurses have to leave their workplace and come in the Health and Occupational Medicine department. Probably some of them didn’t participate because of fear avoidance of clinical examination and/or interview”.
g) authors did not elaborate sufficiently what their findings might offer for the management : a model proposal is added with figure 1 in section “ Conclusions and implications for clinical practice”(lines 386-391).
Doctor Dominique Tripodi and all authors thank very much reviewer 2 for his helpful advice.
